# Multiparametric Whole-Body MRI: A Game Changer in Metastatic Prostate Cancer

**DOI:** 10.3390/cancers16142531

**Published:** 2024-07-13

**Authors:** Arrigo Cattabriga, Benedetta Renzetti, Francesco Galuppi, Laura Bartalena, Caterina Gaudiano, Stefano Brocchi, Alice Rossi, Riccardo Schiavina, Lorenzo Bianchi, Eugenio Brunocilla, Luca Spinozzi, Calogero Catanzaro, Paolo Castellucci, Andrea Farolfi, Stefano Fanti, Nina Tunariu, Cristina Mosconi

**Affiliations:** 1Department of Radiology, IRCCS Azienda Ospedaliero Universitaria di Bologna, 40138 Bologna, Italy; benedetta.renzetti@studio.unibo.it (B.R.); francesco.galuppi@studio.unibo.it (F.G.); laura.bartalena@studio.unibo.it (L.B.); caterina.gaudiano@aosp.bo.it (C.G.); stefano.brocchi@aosp.bo.it (S.B.); cristina.mosconi@aosp.bo.it (C.M.); 2Department of Medical and Surgical Sciences (DIMEC), University of Bologna, 40136 Bologna, Italy; riccardo.schiavina@unibo.it (R.S.); lorenzo.bianchi13@unibo.it (L.B.); eugenio.brunocilla@unibo.it (E.B.); luca.spinozzi2@studio.unibo.it (L.S.); calogero.catanzaro@studio.unibo.it (C.C.); s.fanti@unibo.it (S.F.); 3Radiology Unit, IRCCS Istituto Romagnolo per lo Studio dei Tumori (IRST) “Dino Amadori”, 47014 Meldola, Italy; alice.rossi@irst.emr.it; 4Division of Urology, IRCCS Azienda Ospedaliero Universitaria di Bologna, 40138 Bologna, Italy; 5Nuclear Medicine, IRCCS Azienda Ospedaliero-Universitaria di Bologna, 40138 Bologna, Italy; paolo.castellucci@aosp.bo.it (P.C.); andrea.farolfi@aosp.bo.it (A.F.); 6Clinical Radiology, Royal Marsden Hospital & Institute of Cancer Research, London SW3 6JJ, UK; nina.tunariu@icr.ac.uk

**Keywords:** metastatic prostate cancer, prostate cancer, whole-body MRI, MRI, radiology, oncological radiology

## Abstract

**Simple Summary:**

In the realm of next-generation imaging, whole-body MRI (WB-MRI) is evolving as a key player in the modern management of patients with prostate cancer, showing great potential in the initial staging of high-risk disease, post-treatment evaluation, and recurrence assessment. This technique showcases both notable agreement and complementarity with PET/CT, particularly in identifying secondary bone, nodal, and visceral lesions. WB-MRI’s strength lies in its “one size fits all” nature, avoiding contrast agents and radiotracers and adapting to diverse patient needs. Leveraging DWI and rFF%, it excels in bone metastasis detection, offering a comprehensive view. With promising potential for precise response assessment, WB-MRI emerges as a transformative tool in the management of metastatic prostate cancer.

**Abstract:**

Prostate cancer ranks among the most prevalent tumours globally. While early detection reduces the likelihood of metastasis, managing advanced cases poses challenges in diagnosis and treatment. Current international guidelines support the concurrent use of ^99^Tc-Bone Scintigraphy and Contrast-Enhanced Chest and Abdomen CT for the staging of metastatic disease and response assessment. However, emerging evidence underscores the superiority of next-generation imaging techniques including PSMA-PET/CT and whole-body MRI (WB-MRI). This review explores the relevant scientific literature on the role of WB-MRI in metastatic prostate cancer. This multiparametric imaging technique, combining the high anatomical resolution of standard MRI sequences with functional sequences such as diffusion-weighted imaging (DWI) and bone marrow relative fat fraction (rFF%) has proved effective in comprehensive patient assessment, evaluating local disease, most of the nodal involvement, bone metastases and their complications, and detecting the increasing visceral metastases in prostate cancer. It does have the advantage of avoiding the injection of contrast medium/radionuclide administration, spares the patient the exposure to ionizing radiation, and lacks the confounder of FLARE described with nuclear medicine techniques. Up-to-date literature regarding the diagnostic capabilities of WB-MRI, though still limited compared to PSMA-PET/CT, strongly supports its widespread incorporation into standard clinical practice, alongside the latest nuclear medicine techniques.

## 1. Introduction

Prostate cancer (PC) is the third most common neoplasm worldwide and the fifth leading cause of cancer-related death in men, according to GLOBOCAN 2020, with over 1.4 million new cases and more than 375,000 deaths in 2020 [1,2]. In the United States, one in eight men are diagnosed with PC in their lifetimes. It often presents as a multifocal disease, characterised by high pathological and clinical heterogeneity, with both indolent and highly aggressive forms [3].

The percentage of PC patients who are diagnosed as metastatic can vary depending on several factors, including geographic region, age, and other demographic factors. In general, at the time of diagnosis, most cases are localised, meaning that the disease has not spread beyond the prostate gland. However, about 10–15% of PCs are diagnosed as locally advanced or metastatic [1,4].

The most common sites of extraprostatic dissemination are pelvic lymph nodes, retroperitoneal lymph nodes, and bone, with bone metastases being diagnosed in approximately 80% of advanced PC patients [5,6].

The European Association of Urology (EAU) classifies patients with non-metastatic PCs in four categories (low, intermediate, high risk, and locally advanced disease) that express the likelihood of biochemical recurrence based on cancer grade (i.e., Gleason score) and stage [7]. This classification considers the TNM staging, the ISUP score, and the PSA value at diagnosis.

Multiparametric MRI of the prostate (mpMRI-prostate) combined with clinical parameters has become the gold standard for staging, surpassing the digital rectal examination, being proved to significantly impact the surgical approach in terms of preservation or resection of neurovascular bundles with the aim to reduce positive surgical margins [8].

According to the European Society for Medical Oncology (ESMO), men with localised disease, categorised as having low-risk and favourable intermediate-risk PC, do not need additional imaging, while those with unfavourable intermediate- or high-risk disease should be assessed for nodal and distant metastases [9]. This can be performed with a variety of radiological and nuclear medicine examinations.

In this scenario, whole-body MRI, a novel multiparametric imaging technique, is heralding promising results.

WB-MRI is a next-generation imaging technique standardized with the first version of MET-RADs-P guidelines, published by Padhani et al. in European Urology in 2017 [10]. Even if the excellent diagnostic performances of this technique have been widely demonstrated in PC [11] as well as in other types of diseases (for example, WB-MRI has already been included in the latest version of the International Myeloma Working Group recommendations for Multiple Myeloma), it has not been included yet in the main guidelines for the management of patients with PC, due to lacking level I-III evidence supporting the use of this technique in clinical practice.

This narrative review aims to analyse the main scientific evidence available in the literature, explaining what WB MRI is and showing its decisive role in metastatic PC.

## 2. Methodology

The literature analysis was carried out jointly by AC, BR, and FG. Literature was extracted through the PubMed search engine, following PRISMA checklist, using the keywords “Whole Body MRI” or “Diffusion Weighted Imaging” or “PET-CT and WB-MRI” and “Metastatic Prostate Cancer” and/or “Biochemical Recurrence in Metastatic Prostate Cancer” and/or “DWIBS” and/or “Multiparametric MRI” and/or “response assessment in metastatic prostate cancer” and/or “Dixon Method”, “bone biopsy in metastatic prostate cancer”, “bone metastases in prostate cancer”, “MET-RADs”.

We limited the analysis from 1995 to April 2024. The research produced approximately 1990 articles. After discarding duplicates, 394 articles remained to be analysed. After reading titles and abstracts, 238 papers were excluded as they were considered repetitive, irrelevant, or unrelated. A total of 156 articles were extensively analysed, and 52 were excluded with the following reasons: repetitive, irrelevant, unrelated, out-of-date guidelines, and case reports. Eventually, 104 papers were considered eligible for the purpose of this review. One additional paper was added during the peer-review process (Figure 1).

Given the narrative nature of this review, during the eligibility assessment, the authors particularly favoured international guidelines and papers produced by eminent researchers known for their high expertise in the field. In this setting, no standard appraisal tools were used for quality assessment.

## 3. Discussion

### 3.1. Guidelines in Patients with PC

Despite significant interest in the literature regarding the potential role of WB MRI in patients with PC, the role of this technique is still marginal or not even included in the most widely accepted guidelines for staging and follow-up for PC.

Both European Society for Medical Oncology (ESMO) and European Association of Urology (EAU)–European Association of Nuclear Medicine (EANM)–European Society for Radiotherapy and Oncology (ESTRO)–European Society of Urogenital Radiology (ESUR)–International Society of Geriatric Oncology (SIOG) guidelines agree that patients with localised disease, categorised as having low-risk and favourable intermediate-risk PC (i.e., T1-2a and ISUP 1 or 2 and PSA ≤ 10), do not need additional imaging, while those with unfavourable intermediate- (i.e., T2b and ISUP 3 and/or PSA 10–20) or high-risk disease (≥T2c or ISUP 4–5 or PSA > 20) should undergo metastatic screening for nodal or metastatic disease with conventional imaging and nuclear medicine imaging [7,9].

In this last setting, ESMO guidelines [9] suggest that patients with ISUP 3 intermediate-risk and high-risk disease should be staged for metastases using conventional MRI or CT (abdomen and pelvis) and ^99^Tc Bone Scintigraphy (BS). Next-generation imaging (NGI) techniques such as whole-body MRI and PSMA-PET/CT are taken into account to increase diagnostic performance in high-risk patients together with conventional imaging; however, no sufficient evidence is available to implement in clinical practice yet, since survival benefit due to treatment changes base on NGI is unknown

Indeed, EAU-EANM-ESTRO-ESUR-SIOG Guidelines 2024 highlight the potential of NGI techniques for metastatic screening in patients with high-risk PC [7]. Specifically, in the context of detecting bone metastases, these guidelines refer to studies demonstrating that WB-MRI exhibits higher sensitivity and specificity than a combination of BS and targeted conventional radiography for the detection of bone metastases and similar sensitivity compared to CT scans for the detection of enlarged lymph nodes [13,14]. The imaging modalities recommended for metastatic screening of high-risk PC include PSMA PET/CT (if available) and at least cross-sectional abdominopelvic imaging and BS, since PSMA PET/CT showed significantly higher diagnostic accuracy for staging high-risk disease compared to conventional imaging. However, results from randomised controlled trials (RCTs) evaluating the management and outcomes of patients with (and without) metastases detected by more sensitive imaging are needed before evidence-based decisions can be made.

Regarding metastatic patients, the third version of the Prostate Cancer Working Group Criteria (PCWG3) represents the main accepted criteria for staging and response assessment [15] in patients with castration-resistant prostate cancer (CRPC). According to PCWG3, similarly to the aforementioned guidelines, patients should be assessed with a combination of contrast-enhanced CT thorax–abdomen–pelvis (or, for those intolerant to contrast media, a cross-sectional magnetic resonance imaging scan of the abdomen and pelvis, with an unenhanced CT scan of the chest) and BS. In this case, as well, despite the recognition within the PCWG3 that alternative imaging modalities may offer additional information, particularly in the context of bone metastases, WB-MRI and PET-CT are not recognized imaging modalities in the evaluation of staging and treatment response in metastatic PC, and the recommendation is that these imaging techniques should undergo independent validation to establish their reliability and accuracy before being widely adopted or considered standard practice.

### 3.2. History of Whole-Body MRI

From the 1990s, the use of MRI imaging focused on the bone marrow via conventional T1-weighted and STIR sequences was investigated as an alternative to existing methods for the evaluation of metastatic bone disease [16,17,18,19]. Despite showing greater sensitivity, specificity, and accuracy in the detection of bone lesions compared to BS and radiological skeletal survey [20,21], WB-MRI was not adopted widely due to technical limits such as long examination times and the difficulty in assessing the large quantity of images involved [22]. Since then, advances in MRI technology have allowed for easier performance of WB-MRI examinations. These include, among others, improvements in coil design, parallel imaging, allowing for shorter examination times [23], and continuously moving table, which allows for wider fields of view [24,25], but also advances in dedicated workstation software, such as “stitching” functions, which unify the different stations acquired in WB-MRI in one single stack to facilitate reading [22]. WB-MRI protocols, initially limited to anatomic sequences such as T1-weighted and STIR, were later integrated by diffusion-weighted imaging (DWI). The rationale behind extracranial DWI in oncology rests on the correlation between the restriction of water diffusion and cellularity, providing contrast between highly cellular neoplastic lesions and the less cellular background [26]. Use of whole-body diffusion-weighted sequences with background fat suppression for bone tumours was pioneered in 2004 by Takahara et al. [27], adding functional information to the existing anatomic sequences and becoming the mainstay of current WB-MRI protocols.

### 3.3. Technique and Protocol of Whole-Body MRI

Whole-body MRI protocols as defined by MET-RADS-P [10] are a combination of anatomical and functional sequences.

Core anatomical sequences include both T1- and T2-weighted images.

Most protocols make use of gradient echo (GRE) sequences for T1-weighted images. This allows for shorter acquisition times in comparison to fast spin echo sequences and the possibility of acquiring 3D images. Furthermore, an advantage of GRE acquisition is the possibility of producing multiple image contrasts within a single acquisition through the Dixon method, including in-phase, opposed-phase, fat-only, and water-only images. Fat-fraction (rFF%) maps should be extrapolated by fat-only and water-only images via the formula “rFF% = (F/(F + W)) × 100”. Fat-fraction maps allow for a quantitative measure of fat within lesions, representing a further diagnostic aid when characterizing lesions based on fat content, especially within the bone marrow but also in other organs [28,29,30,31].

T2-weighted imaging, acquired through single-shot or half-acquisition turbo spin echo (HASTE), is useful in the detection of extramedullary disease and in evaluating spinal cord compression [31,32,33].

Sagittal sequences centred on the spine are part of the core WB-MRI protocol; these include T1-weighted images and STIR or fat-suppressed T2-weighted images and are useful in the detection of vertebral lesions, vertebral fractures, and spinal cord compression [10].

The main functional sequence of WB-MRI is represented by whole-body DWI with background body signal suppression (DWIBS), using a single-shot diffusion-weighted echo-planar imaging sequence. DWI is essential to WB-MRI due to its ability to detect highly cellular lesions. For DWIBS, STIR is preferred over other methods such as CHESS for more uniform fat suppression because of its lower sensitivity to magnetic-field inhomogeneities [27]. Usually, multiple averages of DWI data are acquired during free breathing in order to reduce motion artefacts and increase the signal-to-noise ratio [34]. Two b-values are needed for generating apparent diffusion coefficient (ADC) maps for lesion characterization and response assessment [31], the lowest no lower than 50 s/mm^2^ in order to reduce perfusion-related signals and the highest between 800 and 1000 s/mm^2^ [22].

Extensions of the core protocol, such as dedicated brain imaging and MRI mammography, can be implemented in specific cases. The administration of contrast agents may be required for the assessment of specific regions [28].

In patients undergoing WB-MRI for staging of intermediate- to high-risk PC, high-resolution axial T2-weighted and DWI sequences targeted to the prostate can be added to provide an “all-in-one” PC staging examination [35,36].

Mirroring the practice established for PET and CT, WB-MRI is acquired from the vertex to mid-thigh. Acquisition should be extended to the knees in patients with multiple myeloma, and to the feet in cases where the extremities are a possible site of disease, such as neurofibromatosis and Li Fraumeni Syndrome [37,38,39].

Both coronal and axial images have been used in WB-MRI. While coronal acquisition can reduce scan times, it may suffer from more distortion with the same FOV and number of slices [40]. Moreover, images acquired in the axial orientation can be directly correlated with the conventional cross-sectional anatomy of other modalities such as CT [31]. For these reasons, axial acquisition should be preferred.

Slice thickness (SLT) should be the same across the different sequences of WB-MRI in order to facilitate image comparison. The choice of SLT may vary, but the overall recommendation is for the use of contiguous slices with a thickness between 5 mm and 7 mm [10,39].

WB-MRI can be performed on 1.5 T and 3 T scanners, but 1.5 T may be preferred in patients with non-removable metallic prostheses in order to limit susceptibility artefacts and image distortion [28]. Further, 3 T scanners have higher signal-to-noise ratios, but lower field homogeneity may lead to less effective fat suppression and a variable SNR in DWI, increasing the risk of “phase-wrapping” in DIXON scans [22].

Following the protocol described above, WB-MRI can be carried out within approximately 30–60 min depending on the scanner [41,42,43].

Even though it may seem a relatively long acquisition time (most of the oncological MRI examinations take 20–40 min), for a diagnostic examination in cancer patients, prone to the development of painful bone metastases, it has been demonstrated that this technique is well tolerated by such patients [44]; patients and clinicians are advised to ensure optimised pain control prior to scanning.

Some of the main diagnostic sequences in WB-MRI are obtained in the post-processing phase. In order to simplify reading and reporting, images acquired at different stations are composed into a single stack with whole-body coverage. Radial maximum-intensity projections reconstructed from high b-value images are commonly used for navigation and “at-a-glance” detection across the body. Typically, these are displayed in inverse grey-scale to produce a PET-like view [28,31].

Relative fat-fraction maps (rFF%) are obtained from T1-gradient echo Dixon images and give a quantitative assessment of fat distribution in the body and, in particular, bone marrow substitution [45].

ADC maps are necessary for characterising lesions and evaluating the response to therapy [10,39]. ADC is usually calculated with a monoexponential fitting to the signal intensities in the different b-value DWI sequences [46].

### 3.4. Imaging Features of Metastatic PC in WB-MRI

Next-generation imaging (NGI) techniques, such as WB-MRI and PSMA-PET/CT, are emerging in the management of patients with PC [11,47,48].

These methods find application in the initial staging of intermediate- to high-risk PC, post-treatment response assessment of metastatic disease, and staging for patients experiencing biochemical recurrence (BCR) [11].

Comparing the diagnostic performance of WB-MRI with PSMA-PET/CT, several studies demonstrate significant agreement between the two methods in identifying secondary bone, nodal, and visceral lesions, with WB-MRI’s main weakness being represented by the identification of small (less than 8 mm) pathological lymph nodes, a task in which PET shows greater sensitivity [47,48].

Further studies have underlined how the ability to identify metastases with WB-MRI is not significantly influenced by the expertise of the reader. For example, a study by Pricolo et al. evaluated the agreement between two readers with different expertise in interpreting WB-MRI using the MET-RADs-P guidelines. It was demonstrated that the agreement in assessing the response to treatment with WB-MRI was substantial to excellent for bone and retroperitoneal nodal disease (K = 0.61–1), while it was moderate for pelvic nodal disease (K = 0.56). The two readers also demonstrated high agreement in the detection of metastatic sites in primary staging [47].

WB-MRI lends itself to becoming a “one size fits all” solution [10], with strengths, including the absence of contrast agents or radiopharmaceuticals, the lack of specific preparation, and few contraindications other than the presence of incompatible devices or intolerance in claustrophobic patients [49,50].

In comparison, PET also provides functional evaluation but requires specific radiopharmaceuticals tailored to the tumour type, whereas WB-MRI allows for the identification of metastases regardless of histological subtype or the use of radiopharmaceutical drugs. Moreover, up to 10% of PC may not overexpress PSMA, resulting in false-negative PSMA PET/CT in different clinical scenarios [51]. This becomes crucial in the context of metastatic cancer subjected to androgen deprivation, where different neoplastic clones may exhibit varying affinity for radiotracers. In such cases, WB-MRI enables the monitoring of the disease independently from its heterogeneity [47,49].

#### 3.4.1. Evaluation of Bone Metastases with WB-MRI

As already mentioned, according to guidelines, patients should be assessed with a combination of contrast-enhanced CT and BS, CT scan evaluating lymphadenopathies, pathologic soft tissue associated to bone lesions, and visceral metastases, according to RECIST 1.1 criteria and BS evaluating bone lesions [52].

Even if BS is the mainstay of bone metastatic disease evaluation, it is, nevertheless, affected by several limitations. Firstly, BS does not directly evaluate malignant bone disease, rather highlighting reactive osteoblastic uptake. For this reason, it cannot evaluate soft tissue disease and pure lytic metastases, not necessarily reflecting the full burden of disease within the marrow space [53]. Secondly, BS is affected by the so-called FLARE phenomenon; an increase in the number and size of lesions can occur both due to osteoblastic healing associated with tumour response and osteoblastic progression linked to a growing tumour burden, leading to ambiguity when distinguishing between response and progression. To overcome this limit, a confirmatory BS is required to attest the real progression. Thirdly, there are no specific BS criteria for determining progression unless distinct new lesions appear. Enlargement of previously existing lesions is not considered indicative of disease progression. Finally, there are no BS criteria to quantify therapy effectiveness in bone metastases [43].

On the other hand, while CT scan is generally considered a good examination for assessing the presence of enlarged pathologic lymph nodes and visceral metastases, it should not be used to evaluate active bone metastatic disease. It is a common thought that PC bone metastases are almost always sclerotic (i.e., osteoblastic) [54]. Although it is indeed true that PC usually leads to osteoblastic bone metastatic deposits, it can also manifest with osteolytic or mixed-type metastases. Therefore, CT presentation of bone metastases may be rather heterogeneous, ranging from pure lytic lesions with or without soft tissue to ill-defined deposits, poorly distinguishable from background bone marrow on CT, and eventually to “ground-glass” or properly sclerotic deposits, sometimes with new bone formation [55,56].

In 2004, Hamoaka et al. proposed CT criteria to assess the likelihood of response to treatment in bone lesions in patients with bone metastatic breast cancer [57]. According to these criteria, post-treatment sclerosis or a decrease in blastic lesions could be considered reliable features to establish the response to treatment. However, given the fact that many active bone metastases in PC present as sclerotic and that there are no univocal and accepted thresholds to assess the degree of response based on the degree of sclerosis, CT density should not be employed as an imaging biomarker to quantitatively assess the response to treatment in PC patients [10,58,59]. Indeed, sclerotic metastatic lesions maintain their sclerotic appearance, even in the absence of cancer cells or if further sclerosis occurs after treatment. Consequently, it is not possible to reliably assess whether a lesion remains viable or not when comparing images before and after treatment, and CT scan cannot easily distinguish between osteoblastic healing and osteoblastic progression [60].

The significant limitations of these techniques in evaluating bone lesions underline the need to modernise the imaging approach in bone metastatic PC.

By its nature, MRI is one of the best imaging techniques for imaging the bone marrow, being able to visualize both its water and fat content. In the past few years, it has been investigated whether morphological T1- and T2-based sequences would have been of use in the study of metastatic PC patients. For example, one study performed in 2022 by Rydh et al. showed that in patients with PC and PSA < 20 ng/mL, MR imaging of the axial skeleton, performed with sagittal and coronal T1 and STIR sequences, could be more sensitive to bone metastases compared to BS [61].

In 2012, a pioneering study produced by Lecouvet et al. demonstrated the superiority of WB-MRI over a combination of BS and targeted X-ray for the detection of bone metastases and its similar performance compared to CT scan for the detection of enlarged nodes in high-risk PC patients [13]. Two years later, a paper from Pasoglou et al. found similar results with a smaller cohort of patients [14]. Their results are summarized in Table 1. It should be noted that in these studies, WB-MRI was performed using a slightly simplified protocol compared to the modern version proposed by MET-RADs-P.

As previously stated, WB-MRI is a multiparametric technique based on DWI-based, Dixon-based, and other morphological sequences. As expressed by MET-RADs-P guidelines, the information derived from these sequences should always be integrated to avoid misinterpretation of the findings.

In general, lesions that show high signal intensity (SI) in high b-value DWI sequences, low-intermediate ADC values, low signal intensity in rFF% and T1w sequences are highly suspicious for highly cellular, active, bone metastatic deposits (Figure 2).

Bone marrow DWI signal intensity should be compared to that of adjacent muscles to assess whether a bone lesion is characterised by restricted diffusion or not. The degree of restriction is quantitatively expressed by ADC maps but can be measured only when an increased signal intensity is appreciable in both high and low b-value sequences.

The great potential of DWI sequences for assessing the bone marrow (as well as soft tissue) in oncologic patients was introduced in 2004 by Takahara et al. and subsequently exhaustively confirmed by several studies [26,27,62,63,64]. The relevance of quantitatively assessing ADC values has been demonstrated as well, and, while it is not possible to express a single and univocal threshold for this parameter, also due to the variability linked to the choice of the b values, it has been demonstrated that lesions with intermediate to low values, either in the bone marrow or in the soft tissues, are more likely to be malignant [65]. In particular, MET-RADs guidelines state that viable lesions are characterised by ADC values lying between 600–700 μm^2^/s and 1400 μm^2^/s.

In a paper published in 2021, Donners et al. retrospectively analysed the imaging features of a cohort of oncologic patients (mostly affected by metastatic PC), who underwent both multiparametric MRI and subsequent bone marrow CT-guided biopsy within three months. In this study, multiparametric MRI served as a planning imaging technique performed prior to the biopsy; its protocol largely overlapped with that of WB-MRI and was mainly composed of DWI and T1-Dixon sequences. The authors correlated the MRI characteristics of the biopsied lesions with the histological findings, demonstrating that lesions with a high DWI signal, ADC values < 1100 μm^2^/s and rFF% < 20% had higher chance to be positive to biopsy and that multiparametric MRI could identify tumour-positive biopsy with high sensitivity and specificity (respectively, 80% and 82%) and a positive predictive value of 93% [66]. DWI, ADC, and rFF% by themselves had a lower accuracy to identify biopsy-positive lesions, while their combination proved to be effective, underlying the intrinsic multiparametric nature of this technique.

Therefore, it is possible to confidently affirm that, in patients with metastatic PC, bone marrow lesions with apparent diffusion coefficient (ADC) values ranging from 600 to 700 μm^2^/s to 1100 μm^2^/s have a very high chance of representing active metastatic deposits, provided that the other parameters of whole-body MRI are not in contrast.

As an added value, it was demonstrated by Yamamoto et al. that the volume of disease measured on ADC maps in WB-MRI has prognostic value in patients with metastatic PC [67].

The second parameter of WB-MRI is represented by T1-Dixon-based sequences and, in particular, rFF%, as it has been demonstrated that bone metastatic deposits are characterised by significantly low rFF% values compared to normal bone [66,68].

Castagnoli et al. evaluated the WB-MRIs performed on a heterogeneous cohort of 110 patients affected either by multiple myeloma or bone metastatic prostate and breast cancer and demonstrated that malignant bone lesions are characterised by median rFF% values of 13.87%, while normal bone marrow shows significantly higher values (89.76%).

The value of the analysis of bone marrow fat content in distinguishing benign and malignant disease was also demonstrated in 2016 by Yoo et al. [69], who analysed the fat fraction measured on modified Dixon sequences in a heterogeneous cohort composed of healthy patients, patients with benign degenerative disease (i.e., Modic changes, Schmorl’s nodes, benign fractures, etc.), and patients with malignant metastatic disease, demonstrating that rFF% may differentiate between the benign and malignant origins of focal bone marrow abnormalities in situations where the qualitative interpretation of conventional MR images presents challenges.

Donners, Obmann et al. produced a similar paper, retrospectively analysing the performance of both ADC maps and rFF% maps in distinguishing porotic fractures from malignant lesions. They demonstrated that malignant lesions were characterised by significantly lower ADC and rFF% values compared to benign fractures, with the cut-off for the fat fraction being 11.5% and that of ADC being 1040 μm^2^/s. Moreover, rFF% proved to be a better discriminator compared to ADC [70].

In conclusion, all these papers underline the value of rFF% in identifying bone malignancies, and a cut-off of 20% is considered reliable.

Other morphological T1- and T2-weighted sagittal and axial sequences play an important role in evaluating bone lesions, identifying vertebral fractures, metastatic cord compression, epidural invasion, and para-medullary soft tissue. The multiparametric nature of WB-MRI allows for the distinction of porotic and pathologic skeletal-related events.

It is known in the literature that PSMA-PET/CT may be affected by a variable percentage of unspecific bone findings (defined as unspecific bone uptakes, UBUs) that pose a diagnostic challenge. The frequency and body distribution of UBUs vary significantly (0–71.7%) depending on the radiopharmaceutical used, as accurately described by Rizzo et al. [71]. In this context, WB-MRI, due to its multiparametric nature leading to high sensitivity and specificity in bone lesions, may represent a complementary examination capable of resolving doubtful findings detected on PET/CT.

#### 3.4.2. Assessment of Nodal Disease

Lymph node stations are commonly affected by neoplastic infiltration; thus, the early and accurate identification of metastatic nodal disease is crucial for proper patient staging and management. Current guidelines designate CT as the reference standard, offering morphological and dimensional evaluation exclusively, unlike emerging NGI techniques such as WB-MRI and PET-CT, which also provide functional information [50].

Metastatic lymph nodes are most frequently found in the pelvic area, particularly the internal iliac (24%) and obturator fossa (75%), as well as in the retroperitoneum at the para-aortic level (approximately 26%). Hematogenous dissemination is more common in patients with retroperitoneal lymph node involvement [49]. Other lymphatic stations that are less frequently involved include the mediastinum and supraclavicular fossae. A recent study highlights that the distribution of metastatic disease is not influenced by primary treatment [49].

Currently, whole-body MRI employs both dimensional criteria (nodal short axis > 10 mm, or >8 mm for pelvic lymph nodes) and morphological features (including irregular contours, loss of central adipose hilum, or deviation from the normal “kidney shape”) to identify nodal disease [48,49,72] (Figure 3). While MRI exhibits high specificity (86–98%) for identifying lymph node pathology, PSMA and choline PET/CT show greater sensibility [42]. However, studies evaluating the diagnostic performance of choline, Fluciclovine, and PSMA PET/CT to detect nodal involvement using extended pelvic lymph node dissection (ePLND) as the reference standard highlighted the limited diagnostic value (i.e., limited sensitivity) of NGI; thus, ePLND has remained the gold standard for nodal assessment, until now [73,74,75,76].

Since the only MRI criteria to assess nodal disease are size and morphology, the role of DWI as an added parameter has been investigated by several researchers. In particular, ADC values may aid tissue characterization through repeatable measurements [50,72].

ADC values have been proposed as a parameter to differentiate between benign and malignant lymph nodes, especially amongst those appearing morpho-dimensionally normal. Despite several studies showing lower ADC values in lymphadenopathies compared to benign nodes, there is still significant overlap between the two lymph node populations, which prevents a confident distinction [48,49,77,78].

Only a few studies have explored WB-related ADC values, with a recent work focusing on normal lymph nodes in a healthy population, providing specific values for various lymph node stations [50]. For example, the average ADC value of healthy lymph nodes is 1.12 ± 0.27, with the pelvic lymph nodes showing an ADC range between 0.98 and 1.18 and retroperitoneal nodes demonstrating higher average values of 1.61 [50].

Conversely, studies on ADC in pathological lymph nodes predominantly concentrate on nodal disease in the prostatic lodge, measured in multiparametric prostate MRI, consistently demonstrating lower ADC values in metastatic nodes compared to benign ones [11,50,78,79,80,81].

Furthermore, a paper published in 2011 showed inversely proportional correlation between ADC values and the SUV dispersion of PET with choline [80].

Malaspina et al. demonstrated that 74% of metastatic lymph nodes, confirmed by histological examination, had a short axis < 8 mm; the detection rate of nodal metastases for 18F-PSMA PET/CT was 83%, compared to 58% for WB-MRI [82]. Therefore, taking into account that, at present, WB-MRI only considers morphological and dimensional criteria, PSMA-PET/CT provides higher sensitivity and specificity for sub-centimetre lymph nodes. Data are summarized in Table 2.

We are an advocate of the complementary use of WB-MRI and PSMA-PET/CT in this scenario, following multidisciplinary discussions and based on local expertise. If the local practice combines mpMRI-prostate with WB-MRI as initial staging, a PSMA-PET could be performed in cases with no negative distant disease on MRI in order to characterise the potential disease in sub-centimetre nodes. Of course, none of the imaging techniques can replace surgical nodal dissection.

#### 3.4.3. Assessment of Visceral Disease

Visceral metastases are relatively uncommon in patients with advanced PC and were only incorporated in WB-MRI at a later stage by extending the core MRI protocol to include organ evaluation. Some authors argue against total-body coverage, stating that lesions below the knees or extra-vertebral lesions above the diaphragm are rarely identified. However, despite their infrequency, the early identification of visceral metastases is crucial for optimal patient management and to determine the appropriate therapeutic approach, ultimately impacting prognostic outcomes [42,49].

Statistically, patients with a high Gleason score face an increased risk of developing metastases [49].

The most common visceral site of metastasis is the lung, due to hematogenous dissemination through the inferior vena cava or through venous drainage from thoracic vertebral metastases and can affect up to 9% of patients with metastatic CRPC [49].

WB-MRI has limited diagnostic accuracy in evaluating lung parenchyma due to difficulties in identifying sub-centimetre nodules. Therefore, in this regard, CT scan remains the reference imaging modality [11,49].

Another uncommon but prognostically unfavourable site of metastasis is the liver. Hepatic metastases are more frequent in patients with mCRPC, affecting approximately 8.6% of this subgroup and 37% of those with atypical metastatic disease sites. Considering the high choline uptake of healthy liver parenchyma, the known low PSMA uptake of liver metastases, and the unusual cystic appearance of these lesions in CT scan, WB-MRI has proven to represent an excellent imaging technique for identifying liver metastatic lesions. Patients with liver and lung metastases face the worst prognosis, followed by those with bone metastases, irrespective of lymph node involvement, whereas patients with isolated nodal involvement show the best prognosis [49,83].

WB-MRI can additionally identify metastatic lesions in more rare sites, such as adrenal glands, spleen, peritoneum, and brain, as well as recognize complications, such as metastatic cord and cauda equina compression, spinal cord metastases, and hydronephrosis due to retroperitoneal or ureteral involvement [49].

#### 3.4.4. Assessing Local Disease and Biochemical Recurrence

NGI techniques, especially PSMA-PET/CT, show superiority in staging patients with biochemical recurrence, in particular in patients with high-risk BCR [78]. Thus, many factors may influence positive NGI, including PSA levels, PSA doubling time, time of recurrence, ISUP, and pathologic stage as ongoing ADT. Indeed, many tools and nomograms have been developed to predict positive PSMA PET/CT [84,85]. As a consequence, EAU guidelines have recently recognized PSMA-PET/CT as the standard of care in patients with BCR after local treatment (PSA ≥ 0.2 ng/mL after surgery if it would influence subsequent management), while, currently, WB-MRI is not the first-choice method due to limitations in identifying small pathological lymph nodes; however, ongoing studies are aiming to assess its diagnostic accuracy [11].

In 2019, the ST Gallen Advanced Prostate Cancer consensus conference (APCCC) recommended the use of mpMRI combined with PSMA-PET/CT, rather than WB-MRI, in patients with BCR after prostatectomy or radiotherapy (RT) [11].

A prospective study involving 28 patients with BCR after radical prostatectomy demonstrated the diagnostic superiority of PSMA-PET/CT over WB-MRI in identifying metastatic lesions [86]; a further prospective study similarly reported that the addition of WB-MRI to PET/CT did not improve the diagnostic performance [87] while underlining that WB-MRI is more sensitive than reference standards such as bone scintigraphy in the detection of bone metastases [72].

The LOCATE study is currently underway to assess the ability of WB-MRI in detecting nodal and bone metastases in BCR after radiotherapy, and preliminary results suggest inferior performance, if compared to PSMA-PET/CT; however, all these studies have underlined the inadequate diagnostic performance of the methods considered to be the reference standard, such as BS and CT scan [11,78,83].

On the other hand, some studies assessing the combination of prostate mpMRI in association with diffusion-weighted whole-body MRI (with a slightly different protocol compared to the MET-RADs-P standardized protocol) showed promising results, with the added value of achieving a thorough assessment of the patient within a single acquisition [78,88].

A study by Jannusch et al. analysed the diagnostic performance of whole-body PSMA-PET/MRI with dedicated high-resolution MRI sequences on the pelvis for the identification of disease recurrence in 102 patients with biochemical recurrence. The study demonstrated the complementarity between the PSMA-PET component and the MRI component, in particular showing the superiority of MRI over PET/CT for the detection of local recurrence and the superiority of PET/CT over MRI for the detection of metastatic lymph nodes (especially in patients with PSA < 1.69 ng/mL) [89]. The whole-body MRI protocol was composed of DWI and 3D Dixon Vibe sequences (therefore, lacking part of the sequences characterising standard MET-RADS-P WB-MRI protocol). Data from this study suggest that WB-MRI, potentially supplemented by dedicated high-resolution images through the prostatic bed, can offer added value compared to PET/CT for the early detection of local recurrence, especially in patients who have undergone prostatectomy, as it is not affected by the accumulation of radiotracers within the bladder, so this could be an interesting topic for future research (Figure 4).

Overall, the use of NGI compared to standard methods has demonstrated its diagnostic superiority in identifying distant metastases compared to the anatomical sequences of standard imaging alone. According to the present guidelines, NGI techniques are not universally recommended yet; however, their diagnostic potential, especially in specific patient scenarios, makes them valuable tools in the evolving landscape of biochemical recurrence management. In this setting, PSMA-PET/CT seems to show better performances compared to WB-MRI in patients with small nodal recurrence. However, studies with larger cohorts of patients should be carried out [90]. 

### 3.5. Response Assessment in Metastatic Prostate Cancer

Standard imaging techniques, such as CT and BS, have significant limits in the evaluation of the burden of bone metastases and response to treatment [91]. The widely used RECIST v1.1 [52] does not define response in bone lesions, as they are considered non-measurable diseases. The PCWG3 defines progression as the appearance of new lesions on BS but does not define criteria for treatment response [15]. Therefore, the tumour response in patients with bone-only metastatic disease relies solely on a decrease in the PSA level, which has not been proven to be a surrogate marker for survival [92,93].

Hence, other imaging modalities have been investigated as response biomarkers, including WB-MRI. Overall, the pattern of bone marrow infiltration by metastatic disease can be focal or diffuse, and subsequent MRI imaging after therapy may show evolution of these patterns [94].

As extensively described by Lecouvet et al. in a review published in 2013, several morphological features can be used to assess response or progression after treatment (see Table 3). In particular, the appearance of new focal lesions in previously normal bone marrow or an increase in the number/size of already existing lesions or the evolution from focal to diffuse infiltration are features indicative of progressive disease [94]. Conversely, the disappearance of focal lesions and/or their decrease in the number/size or the return of normal fatty marrow from a previously diffuse pattern are indicative of response. The total disappearance of focal lesions and diffuse infiltration corresponds to complete “imaging” remission.

Stability in the size and morphology of lesions on anatomical sequences does not have a clear meaning; persistent bone marrow abnormalities may represent partially responsive disease or completely treated lesions with residual necrosis or fibrosis.

Skeletal-related events (SREs), defined as asymptomatic nonclinical fractures on serial imaging, clinical pathologic fractures, or spinal cord compression, are considered signs of progression [15].

In addition to the size and number of lesions, a few specific morphological signs can help to characterise the response to treatment of individual lesions.

The appearance of a peripheral halo of high signal intensity on T1-weighted images (“fatty halo sign”) indicates response. Conversely, the appearance of a hyperintense halo in STIR or T2-weighted fat-saturated images (“cellular halo sign”) is indicative of activity or progression. The disappearance of the cellular halo is considered an early sign of response.

Soft tissue extension is also a fundamental parameter of response: an increase in or the appearance of soft tissue extension is a sign of disease progression, and para-medullary extension must be especially highlighted in cases of involvement of the vertebral canal. On the other hand, a decrease in or the disappearance of soft tissue involvement indicates a responsive lesion.

Pitfalls in the response assessment may be encountered when a patient is treated with marrow-stimulating factors, leading to alterations in marrow cellularity and a decrease in T1-weighted signal intensity that could be mistaken for progression to diffuse neoplastic infiltration [95,96]. The repetition of MRI weeks after the end of the stimulating treatment shows a return to normal marrow appearance. Another potential pitfall of morphological analysis is a rare “flare phenomenon”, which represents an increase in diameter in T1-weighted images due to post-therapy oedema [97]. Skeletal-related events (SREs) such as vertebral fractures due to underlying neoplastic lesions are usually interpreted as a sign of progression. However, PC patients may experience non-malignant vertebral fractures from unrelated and/or pre-existing conditions such as osteoporosis that cannot be confidently differentiated from malignant fractures based on anatomical sequences only.

Overall, the use of morphological and size criteria in MRI for the response assessment of bone marrow metastatic lesions faces limitations due to the difficult interpretation of persistent lesions after therapy and because new drugs such as cytostatic therapies have a limited impact on the lesion size, despite histological efficacy [98].

Response to treatment of bone lesions must, therefore, also be assessed through the functional information provided by DWI sequences [64], as they can detect variations in water diffusion after therapy, resulting from changes in cellularity and a loss of membrane integrity [62,99]. A decrease in signal intensity on high *b*-value images with a corresponding increase in ADC is considered indicative of response. ADC increases within PC metastases treated with antiandrogen therapy may be detected as early as 1 month after the start of treatment [62,100]. However, lesions that are responding to systemic or radiation therapy may show an increase in signal on high b-value DWI, with a corresponding increase in the ADC value in relation to the “T2 shine through effect”, reflecting an increase in the water content due to necrosis or oedema [98].

Quantitative changes in the ADC value have been investigated as an objective response criterion and are among the main response criteria in the MET-RADS-P recommendations. Several studies have found a significant increase in the ADC of bone marrow lesions in patients responding to therapy [100,101,102,103]. A 2011 study by Messiou et al. found the overall ADC of bone lesions increasing in both responders and progressors. However, the magnitude of the ADC increase was greater in responders, with an increase in the overall ADC of more than 25%, with 75% being sensitive and 66.6% specific for response [101]. An ADC decrease in responding lesions is possible and is thought to be associated with the development of sclerosis or reemergence of fatty marrow within the treated lesion [100,101], underlying the importance of evaluating ADC maps together with anatomical images, especially T1-weighted sequences and their derived rFF% maps.

**Table 3 cancers-16-02531-t003:** Response assessment in metastatic prostate cancer with different techniques. * see original article for details.

Morphological Patterns of Response			
Paper	Technique	Response	Progression
Scher et al. *J Clin Oncol* 2016 (PCWG3) * [16]	CT-scan + Bone Scan	Nodal and visceral disease: as defined according to RECIST 1.1 Bone disease: no BS criteria for assessing response in bone disease	Nodal and visceral disease: as defined according to RECIST 1.1 Bone disease: appearance of new lesions according the “2+2 rule” within the flair period and “2+0 rule” beyond the flare period.
Lecouvet et al. *European Radiology* 2013 [95]	MRI (bone disease)	Disappearance of previously visible focal lesions Decrease in size of existing lesions Return of fatty marrow from a previously diffuse pattern Fatty marrow reemergence within a lesion Appearance of a “fatty halo sign” (peripheral halo of T1 hyperintensity) Disappearance of a “cellular halo sign” Reduction in size/disappearance of previously noted soft tissue associated with a bone lesion	Appearance of new focal lesions Increase in size of previously existing lesions Evolution from focal to diffuse marrow involvement Appearance of a “cellular halo sign” (STIR peripheral STIR hyperintensity) Appearance/increase in size of soft tissue associated to bone lesions Skeletal related events
Multiparametric Patterns of Response			
Padhani et al *Eur Urol* 2017 (MET-RADs-P) * [10]	WB-MRI	Nodal and visceral disease: as defined according to RECIST 1.1 Decrease in number/size of existing lesions Evolution from diffuse to focal pattern Significant decrease in DWI signal intensity with increase in ADC values ≥ 40% ≥1400 μm2/s Fatty marrow re-emergence Disappearance of “hypercellular halo” on T2	Nodal and visceral disease: as defined according to RECIST 1.1 Increase in number/size of focal lesions Evolution from focal to diffuse pattern New areas of focal or diffuse infiltration on previously healthy marrow Appearance/increasing in soft tissue associated with skeletal lesions Appearance of new areas of DWI hyperintensity with pathologic ADC values (600–1000 μm^2^/s)
Messiou et al. *Eur Radiol* 2011 [102]	Diffusion Weighted MRI	Overall ADC of bone lesions increases both in responders and progressors, but the magnitude of increase if higher for responders. An increase in overall ADC > 25% is 75% sensitive and 66.6% specific for response. ADC alone cannot confidently assess response or progression as the changes in bone marrow composition significantly influence ADC values.	

The response assessment of metastatic lesions has been standardised by the MET-RADS-P criteria. This involves recording changes at a regional level and, overall, through dedicated template forms.

The regional response of nodal and visceral disease must be assessed with RECIST criteria v1.1 [52], as modified by the PCWG3.

The regional response of bone disease is scored on a scale of 1 to 5: (1) highly likely to respond, (2) likely to respond, (3) stable, (4) likely to progress, and (5) highly likely to progress [10].

MET-RADS-P includes a scoring system that allows one to record the heterogeneity of responses at the regional level. This is a three-pattern scoring system: the primary or dominant pattern in the region is the one seen in the majority of lesions, and the secondary pattern is the second most frequent. When there are three patterns in a region and the tertiary pattern is RAC 1–3, it can be ignored as a minor response or stable disease and should only be recorded to document a pattern of progressive disease (RAC 4 and 5) occurring in a minority of lesions.

The status of primary disease, nodes, viscera, and bone disease is recorded separately using the overall response assessment template form.

The overall response for primary tumour, nodal, and visceral disease should be assessed via PCWG3 modifications using RECIST v 1.1 [15] and assigned the following categories: complete response, partial response, stable disease, progressive disease, and discordant.

The overall response of bone disease is on a 1-to-5 scale, indicating the likely overall response category: (1) highly likely to respond, (2) likely to respond, (3) stable, (4) likely to progress, and (5) highly likely to progress (Figure 5).

It has been acknowledged that WB-MRI possesses advantages over existing imaging methods of response assessment. In contrast to PET-CT, WB-MRI does not rely on the affinity of tumour cells for a tracer or the presence of receptors, making it a consistent and universal tool for response assessment. While PSMA-PET/CT has proven efficient in biochemical recurrence and newly diagnosed disease, it should be considered with caution for response evaluation, not only for its variable affinity but also because affinity can vary with the line and type of treatment. The interaction between androgen signalling and PSMA expression is complex and still being studied. However, prolonged androgen blockade leads to the downregulation of PSMA expression, reducing the visibility of metastatic disease [104,105]. On the other hand, short-term androgen blockade upregulates PSMA, leading to a flare effect in PC. This phenomenon does not seem to be as transient as that of BS, and it can plateau, leading to misinterpretation as progressive lesions [106].

## 4. Conclusions

WB-MRI is emerging as a promising and versatile tool in managing prostate cancer patients.

The multiparametric nature of WB-MRI, encompassing anatomical sequences with functional diffusion-weighted imaging (DWI) and relative fat fraction (rFF%) sequences, enhances its diagnostic potential. Studies highlight the effectiveness of DWI in assessing bone marrow, the integration of ADC values and rFF% further refining its diagnostic accuracy.

Acknowledging WB-MRI’s challenges in sub-centimetre nodes and lung metastases assessments, WB-MRI shows promising results in detecting bone and visceral disease with high sensitivity and offers a combined assessment of treatment response and prompt disease complication detection. In cases of local and biochemical recurrence, WB-MRI offers valuable complementarity with PSMA-PET assessments, showcasing its potential role in the evolving landscape of prostate cancer management. 

Further research and standardisation will likely solidify its position in PC assessment. Moreover, further efforts should be made to define the complementary use of next-generation imaging techniques, according to individual patient disease characteristics, for a personalized approach.

## Figures and Tables

**Figure 1 cancers-16-02531-f001:**
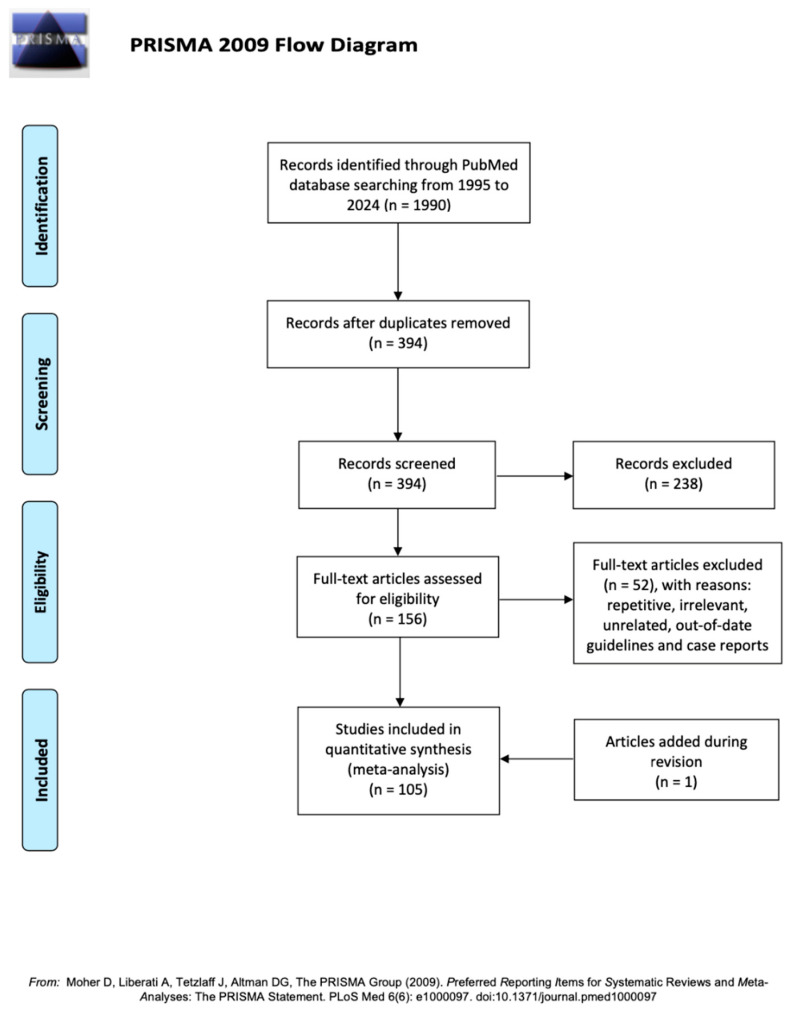
PRISMA flow diagram [12].

**Figure 2 cancers-16-02531-f002:**
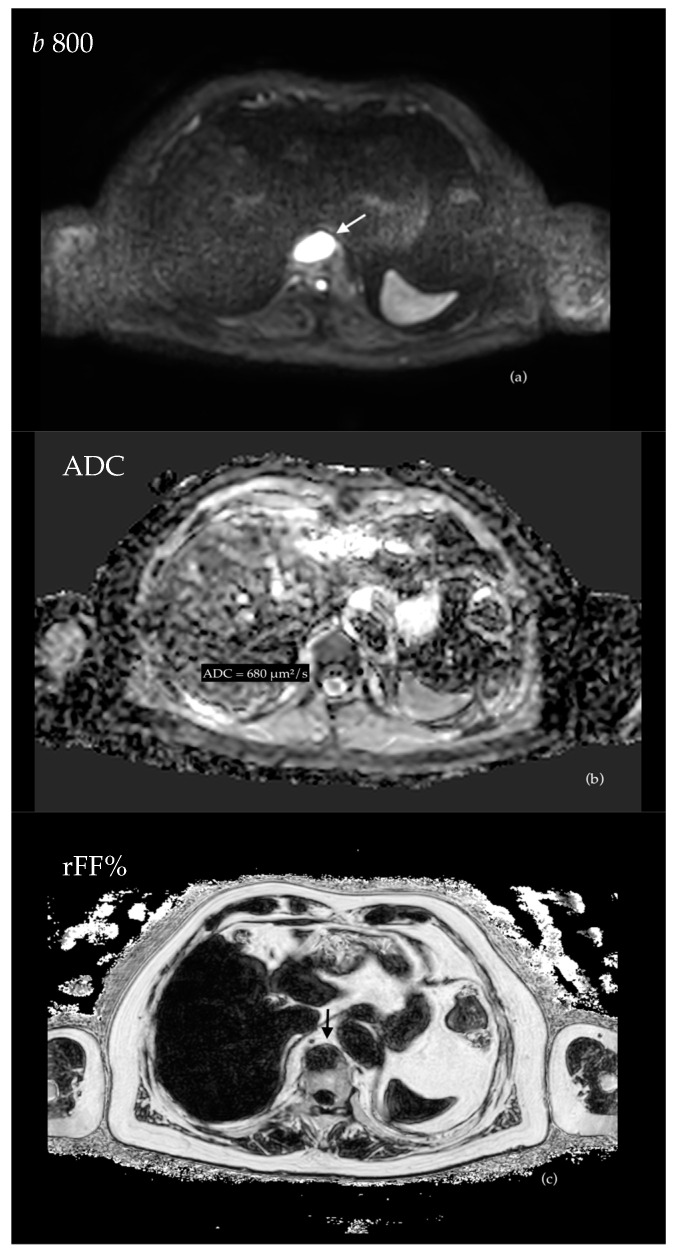
A 64 year-old patient who underwent radical prostatectomy for PC (Gleason score 4 + 4) presented with biochemical recurrence. A single focal bone metastasis located within the anterior aspect of a thoracic vertebral body (arrows) shows high DWI signal intensity (**a**), relatively low ADC values (mean ADC = 680 μm^2^/s) (**b**) and appears markedly hypointense on rFF% (**c**).

**Figure 3 cancers-16-02531-f003:**
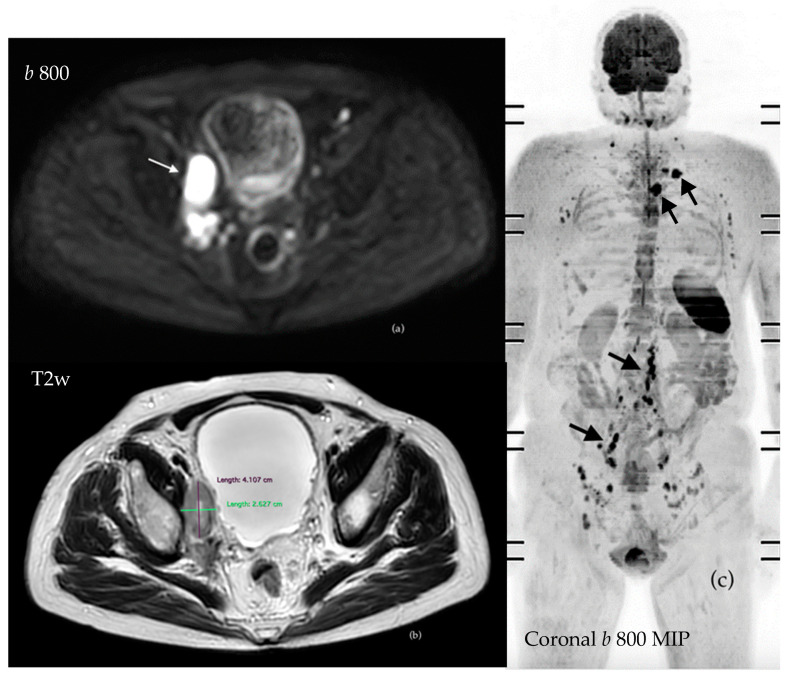
PC patient who had undergone radical prostatectomy in 2019 presented with significant biochemical relapse in 2022. WB-MRI showed supra- and infradiaphragmatic nodal disease (arrows), which can be easily identified through the assessment of DWI images (**a**) and measured on morphological axial T2 sequences (**b**). Three-dimensional MIP offers a panoramic visualization of the disease burden (**c**).

**Figure 4 cancers-16-02531-f004:**
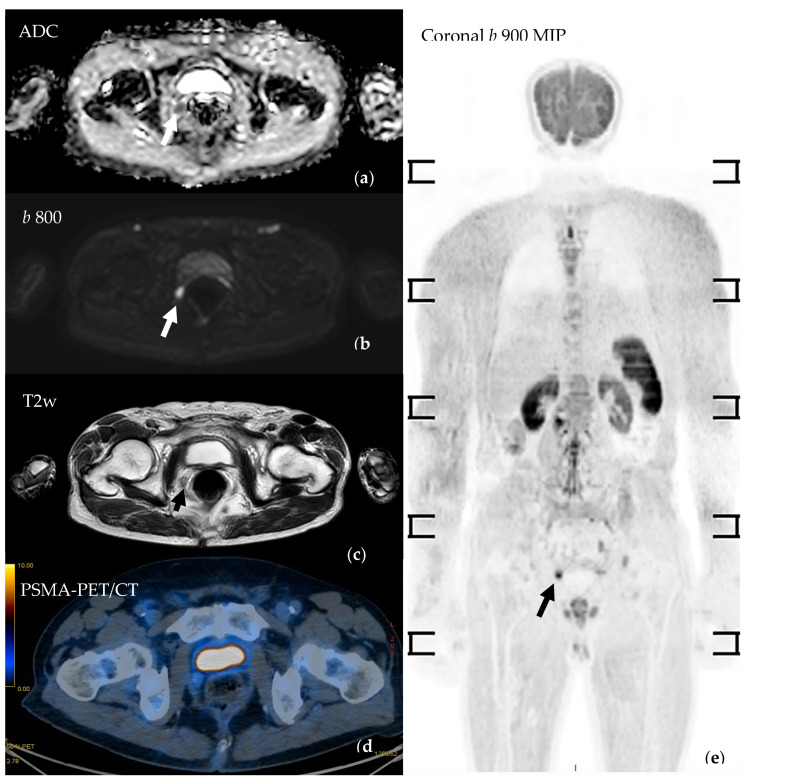
Prostate cancer patient (Gleason score 4 + 4) who underwent robot-assisted radical prostatectomy presented with biochemical recurrence. WB-MRI demonstrated a focal signal abnormality (arrows) consistent with local recurrence. (**a**) Axial b 800 DWIBS shows a focus of impeded diffusion in the right para-rectal space being confirmed in the ADC maps (**b**) and by the axial T2w images (**c**). Coronal maximum-intensity projection of b 800 DWIBS (**e**) offers a panoramic view. PSMA-PET/CT (**d**) could not detect the disease relapse; it was subsequently confirmed by choline-PET/CT.

**Figure 5 cancers-16-02531-f005:**
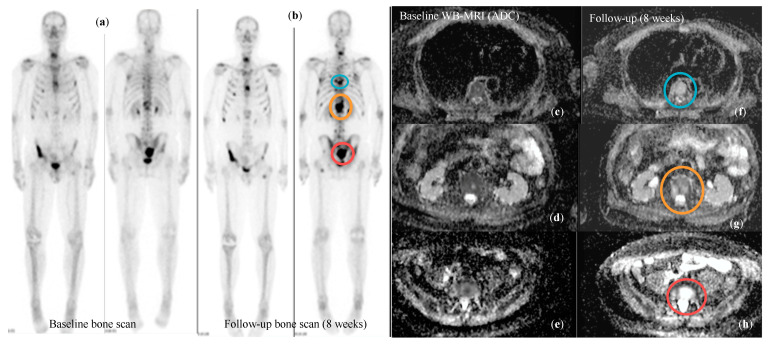
Patient with bone metastatic CRPC who underwent baseline staging with standard techniques (BS and CT scan) and with WB-MRI prior to entering a clinical trial and after 8 weeks of treatment. Follow-up BS (**b**) suggested an increase in the disease burden compared to baseline (**a**). The same lesions showed signal characteristics consistent with highly cellular/active disease on baseline WB-MRI (**c**–**e**) (mean ADC = 833 μm^2^/s), while at follow-up scan (**f**–**h**), there was a significant increase in ADC values, in keeping with response to treatment (mean ADC = 1463 μm^2^/s; RAC = 1). BS appearances were interpreted as FLARE.

**Table 1 cancers-16-02531-t001:** Diagnostic performance of WB-MRI for detecting bone metastases. Data are expressed at the patient level.

Study	Population	WB-MRI		Standard Techniques			Comments/Conclusions
		Sens %	Spec %	Technique	Sens %	Spec %	
Lecouvet et al. *European Radiology* 2012 [14]	100 high risk PCa patients	98–100	98–100	BS ± X-ray	86	98	WB-MRI shows higher sensitivity and similar specificity compared to the combination of BS ± X-ray for the detection of bone metastases
Pasoglou et al. *The Prostate* 2014 [15]	30 high risk PCa patients	100	100	BS ± X-ray	89	90	Comparison of AUC shows no significant difference between WB-MRI & BS ± X-ray. However, WB-MRI is significantly better for Global Metastatic Status (Bone + Node) Sens% = 100 vs. 85
Johnston et al. *Eur Radiol* 2019 [43]	56 PCa patients only 33 ^18^F-choline PET/CT	90	88	BS	60	100	WB-MRI outperforms BS for detecting bone lesions, while is comparable to ^18^F-choline PET/CT.
				^18^F-choline PET/CT	80	92	

**Table 2 cancers-16-02531-t002:** Diagnostic performance of WB-MRI for the detection of nodal disease. Data are expressed at the patient level.

Study	Patients Enrolled	WB-MRI		Compared Technique			Comments/Conclusions
		Sens %	Spec %	Technique	Sens %	Spec %	
Lecouvet et al. *European Radiology* 2012 [14]	100 high risk PCa patients	77–82	96–98	CT-scan	77–82	95–96	WB-MRI and CT-scan show similar performancesfor detecting of enlarged lymph nodes
Pasoglou et al. *The Prostate* 2014 [15]	30 high risk PCa patients	100	100%	CT-scan	82	100	Comparison of AUC shows no significant difference between WB-MRI & CT-scan. However, WB-MRI is significantly better for Global Metastatic Status (Bone + Node) Sens = 100% vs. 85%
Malaspina et al. *Eur J Nucl Med Mol Imaging* 2021 [83]	78 PCa patients	40–50	96–91	PSMA-PET/CT	84–90	94–96	PSMA-PET/CT outperforms WB-MRI for the detection of pathologic lymph nodes. 74% of metastatic lymph nodes, confirmed by histological examination, had a short axis < 8 mm
Johnston et al. *Eur Radiol* 2019 [43]	33 PCa patients	100	96	^18^F-choline PET/CT	100	82	WB-MRI & ^18^F-choline PET/CT have similar diagnostic performance for detecting nodal disease (however small cohort)

## Data Availability

No new data were created or analyzed in this study. Data sharing is not applicable to this article.

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
