# Peer review of "Multiparametric Whole-Body MRI: A Game Changer in Metastatic Prostate Cancer"

_cancers, 2024, doi:10.3390/cancers16142531_

Round 1

Reviewer 1 Report

Comments and Suggestions for Authors

In this review paper, the authors explored relevant scientific literature on the role of WB-MRI in mPC and summarized the major findings in the reported studies. In general, this topic is clinically interesting and important. The authors well collected and summarized the available data, illustrated the results, and comprehensively discussed the findings from the data. The literature review procedure generally followed the PRISMA 2009 statement.

I am generally satisfied with the topic, context, presentation, and writing of the manuscript. But, there are still several issues to be addressed for quality improvement before acceptance. One major concern is the confusion between multi-parametric (pelvic and/or other regions) MRI (mp-MRI) and WB-MRI. Although these two protocols have common sequences and techniques such as DWI and Dixon, I think it may be misleading to present the reported findings in literature based on the presumption that they are equivalent, since their design, implementation and imaging parameters could be substantially different. There are many places in the context to cite the results in multi-parametric (pelvic and/or other regions) MRI studies but claim them to be WB-MRI. For example, reference 65 was not a WB-MRI study but rather a multi-parametric MRI study. Meanwhile, this study included not only PC patients. In addition, the original results in this study showed that the single MRI parameters DWI signal, ADC, and rFF failed to distinguish between tumour-positive and tumour-negative biopsies (each p > 0.082), but this review did not clearly present this negative result, while only showing the positive result by combing these three parameters. I suggest the authors should clearly explain the difference between mpMRI (I do not exactly mean the mpMRI as defined in PI-RADS) and WB-MRI. Another concern is the slight overstatement and some improper citations. The tone is generally too optimistic. But, based on the still limited performance of WB-MRI in some aspects, e.g. metastatic lymph node detection, I don’t think “WB-MRI stands out for its pivotal role in initial staging of high risk disease, post-treatment evaluation and recurrence assessment”. WB-MRI may be showing good performance in mPC staging on the patient basis, in particular in patients with bone and visceral mets, but not on the lesion basis. Thus, in the manuscript, I suggest the authors clearly showing the diagnostic metrics, like sensitivity and specificity, were patient-based or lesion-based. Regarding improper citation, e.g. ref14 is only applicable for mCRPC. No existing clinical guidelines confirmed the role of NGI either. Another example was that “The use of mpMRI of the prostatic lodge is recognized as the gold standard for identifying local recurrence.” Lines 472-473. Actually, I did not find this statement in reference 11. Meanwhile, here indicated mpMRI but not WB-MRI. In lines 499-500, the authors cited ref86 which was for post-prostatectomy, while giving an illustration of a patient with prostate. In section 3.5, most references were not based on WB-MRI except for the proposed MET-RADS-P.

Regarding result description and presentation, it is too description by text. I would suggest authors use some tables to present key information and data for better categorization, clarification, and easier understanding for readers. 

There are also a number of specific comments. In lines 57-58, risk stratification also uses PSA information. Regarding Fig 1, why did the search in a single database lead to so many duplicates? What are the “additional records identified through other sources”? They were not mentioned in the context. Some abbreviations should be explained at their first use. Some abbreviations were not consistent, e.g. rF% and rFF. I would also suggest giving some patient characteristics for Figs. 2-4.  

Comments on the Quality of English Language

It is generally good. But, the authors should carefully proofread the manuscript for language editing too.             

Author Response

Reviewer 1

In this review paper, the authors explored relevant scientific literature on the role of WB-MRI in mPC and summarized the major findings in the reported studies. In general, this topic is clinically interesting and important. The authors well collected and summarized the available data, illustrated the results, and comprehensively discussed the findings from the data. The literature review procedure generally followed the PRISMA 2009 statement.

I am generally satisfied with the topic, context, presentation, and writing of the manuscript. But, there are still several issues to be addressed for quality improvement before acceptance.

One major concern is the confusion between multi-parametric (pelvic and/or other regions) MRI (mp-MRI) and WB-MRI. Although these two protocols have common sequences and techniques such as DWI and Dixon, I think it may be misleading to present the reported findings in literature based on the presumption that they are equivalent, since their design, implementation and imaging parameters could be substantially different. There are many places in the context to cite the results in multi-parametric (pelvic and/or other regions) MRI studies but claim them to be WB-MRI. For example, reference 65 was not a WB-MRI study but rather a multi-parametric MRI study. Meanwhile, this study included not only PC patients. In addition, the original results in this study showed that the single MRI parameters DWI signal, ADC, and rFF failed to distinguish between tumour-positive and tumour-negative biopsies (each p > 0.082), but this review did not clearly present this negative result, while only showing the positive result by combing these three parameters. I suggest the authors should clearly explain the difference between mpMRI (I do not exactly mean the mpMRI as defined in PI-RADS) and WB-MRI.

Firstly, we would like to let you know that we are very grateful for the time you spent reviewing our article. Thanks to your highly pertinent comments and we think that the quality of our paper has significantly improved.

As you correctly pointed out, while multiparametric MRI and WB-MRI share their core protocol (in particular DWI and Dixon sequences) it may be misleading considering the first one as a “shorter” version of the second one, as sometimes they may vary significantly in terms of indication and parameters. In the specific case of the article published by Donners et al. (ref 65) it is our understanding that the parameters used to perform mp-MRI largely overlap with those used to perform WB-MRI in the same institution. Nevertheless, the two techniques may have different indications and, in other contexts, their protocols may be dissimilar. Indeed, in the first version of our paper this difference was not specified, leading to possible misunderstandings for the readers as well as to suboptimal information. Thanks to your wise advice, we had the chance to better specify the difference between the two techniques and clarify in which studies WB-MRI was actually used. Following your suggestion, we also explained how in ref 65 only the combination of ADC, DWI and rF% were able to identify biopsy-positive lesions. Moreover, we specified that not all patients enrolled in that study were prostate cancer patients.

Another concern is the slight overstatement and some improper citations. The tone is generally too optimistic. But, based on the still limited performance of WB-MRI in some aspects, e.g. metastatic lymph node detection, I don’t think “WB-MRI stands out for its pivotal role in initial staging of high risk disease, post-treatment evaluation and recurrence assessment”. WB-MRI may be showing good performance in mPC staging on the patient basis, in particular in patients with bone and visceral mets, but not on the lesion basis. Thus, in the manuscript, I suggest the authors clearly showing the diagnostic metrics, like sensitivity and specificity, were patient-based or lesion-based.

Thank you very much for your kind opinion. Indeed, some sections of this paper have been expressed with an excessively optimistic tone, that could mislead the reader. We modified the text accordingly, trying to adopt a less enthusiastic and more neutral tone. Following your advice, we detailed the diagnostic metrics of the more representative articles within tables, specifying when they were patient based.

Regarding improper citation, e.g. ref14 is only applicable for mCRPC. No existing clinical guidelines confirmed the role of NGI either.

Thank you very much for this annotation. We amended this mistake, explaining that PCWG3 are intended for castration resistant prostate cancer patients.

Another example was that “The use of mpMRI of the prostatic lodge is recognized as the gold standard for identifying local recurrence.” Lines 472-473. Actually, I did not find this statement in reference 11. Meanwhile, here indicated mpMRI but not WB-MRI.

Thank you very much for your punctual annotation. Indeed, there may have been a mistake within this citation. Besides, the first sentence of the paragraph was intended as an introduction stating that even if multiparametric MRI of the prostate has a known value for the assessment of local recurrence, WB-MRI may play a role in this setting. However, on a second thought, we found this sentence rather pleonastic and deleted it.

In lines 499-500, the authors cited ref86 which was for post-prostatectomy, while giving an illustration of a patient with prostate. In section 3.5, most references were not based on WB-MRI except for the proposed MET-RADS-P. 

Thank you for your kind comment. We added some new figures showing a case of local recurrence in a patient who has undergone prostatectomy. It is indeed true that most reference were not based on WB-MRI according to MET-RADs guideline, as the literature on this specific topic is still quite limited. Most of the cited articles referred to Diffusion-Weighted whole Body MRI with morphological T1 and T1 sequences, that differ from the standard protocol mostly due to the lacking of T1-Dixon / Fat Fraction. We think these studies can be considered interesting for our review, as for the identification of local recurrence DWIBS/ADC sequences are the most crucial, while T1-Dixon / Fat Fraction sequences are mostly used for assessing bone metastatic deposits. However, we appreciate that the original version of our paper may have been misleading in this sense, and in this new version we specified that the studies in ref 76 and 81 used Diffusion-Weighted whole-body MRI.

Regarding result description and presentation, it is too description by text. I would suggest authors use some tables to present key information and data for better categorization, clarification, and easier understanding for readers.

We really appreciate your suggestion. Indeed, in the original version of our work, the absence of any tables organizing the information made the paper look dense and less accessible. Following your advice, we have added some tables summarizing the main evidence.

There are also a number of specific comments. In lines 57-58, risk stratification also uses PSA information. Regarding Fig 1, why did the search in a single database lead to so many duplicates? What are the “additional records identified through other sources”? They were not mentioned in the context. Some abbreviations should be explained at their first use. Some abbreviations were not consistent, e.g. rF% and rFF. I would also suggest giving some patient characteristics for Figs. 2-4.  

Thank you very much for these specific comments. We added a sentence explaining which parameters are considered for the EAU risk-stratification of prostate cancer patients. Regarding the large number of duplicates, this is likely caused by the fact that many researched topics are quite general and may lead to overlapping results. The “additional records identified through other sources” equals to 0 in our paper. We deleted the box. We checked the abbreviations and amended when inconsistency was found (such as rFF% vs FF, PC vs PCa etc.).

Finally, we’d like to inform you that, following the other reviewer’s advice, we added two new references, detailing it in the text.

Reviewer 2 Report

Comments and Suggestions for Authors

This is a narrative review on the value of whole body MRI in the staging of prostate cancer. Eventually, 103 papers have been selected and eligible for the purpose of the review. 

The selection of papers was described a bit vague to my opinion. Was there more than one person involved in the selection of these papers? No quality assessment of selected papers was performed. 

The authors need to be complemented on their detailed assessment on the accuracy of MRI as a staging tool. Could the authors clarify the role of WB-MRI in the staging of patients with intended locally confined disease such as those with unfavorable intermediate and high-risk prostate cancer and in those with suspected metastatic disease (i.e. those with initial PSA > 50 ng/mL)?

What is the intraobserver and interobserver variability of WB-MRI? Is this known?

As of today, PSMA PET has become the standard staging module for patients with unfavorable intermediate and high-risk prostate cancer in the majority of western European countries. With this in mind, what is the role of WB-MRI in these patients?

Author Response

Reviewer 2

This is a narrative review on the value of whole body MRI in the staging of prostate cancer. Eventually, 103 papers have been selected and eligible for the purpose of the review. 

The selection of papers was described a bit vague to my opinion. Was there more than one person involved in the selection of these papers? No quality assessment of selected papers was performed.

Firstly, we would like to express our gratitude for the time you spent reviewing our article. Your highly pertinent comments have significantly improved the quality of our paper.
The selection of papers was carried out jointly by three authors, who analyzed the literature and chose the most relevant and pertinent articles. Given the narrative and descriptive nature of our review, a significant part of our paper is based on international guidelines and studies produced by eminent authors. As you correctly pointed out, we did not use critical appraisal tools for evaluating the quality of the studies, but we jointly analyzed the papers to assess their quality and eligibility.
We’d like to specify that, to address your comments, we deemed useful add two new references (we updated the PRISMA WORKFLOW). One of the references were discarded during the eligibility phase, as previously deemed repetitive, but we found it fitting for answer your questions. A second reference was not originally included in our research (also considering that the article has been published in June 2024), but we deemed it useful to explain the complementarity between WBMRI and PSMA PET CT. We specified it within our paper.

The authors need to be complemented on their detailed assessment on the accuracy of MRI as a staging tool. Could the authors clarify the role of WB-MRI in the staging of patients with intended locally confined disease such as those with unfavorable intermediate and high-risk prostate cancer and in those with suspected metastatic disease (i.e. those with initial PSA > 50 ng/mL)?

Thank you for your valid comments which we hope that we have satisfactorily addressed. There is limited evidence of WBMRI as a staging tool for patients with prostate cancer. This is due to the known limitations of WBMRI in detection of subcentimetre nodal involvement. We advocate the complementarity of MRI to PSMA PET and CT and the sequential use of these techniques. For example, should one use WBMRI together with mpMRI prostate for initial staging and the MRI techniques show no significant disease outside prostate than a PSMA PET should be undertaken if radical therapy is considered as it is well known that MRI will underestimate the nodal disease. If the MRI techniques show bone metastases or retroperitoneal pathological nodes than additional PSMA would not change the management. We have added a paragraph to address your suggestion.

What is the intraobserver and interobserver variability of WB-MRI? Is this known?

Thank you very much for this question. Several studies in literature have investigated the inter-observer agreement in reporting WBMRI both for prostate cancer (MET-RADs criteria) and for multiple myeloma patients (MY-RADs criteria) showing high concordance between readers with different expertise. Probably we did not give enough importance to this aspect in the first version of our paper. Following your kind advice, with added a paragraph describing the results of an article about the inter-observer agreement in WB-MRI.

As of today, PSMA PET has become the standard staging module for patients with unfavorable intermediate and high-risk prostate cancer in the majority of western European countries. With this in mind, what is the role of WB-MRI in these patients?

Once gain we advocate the complementarity between PSMA and WBMRI and use of these techniques together following multidisciplinary team discussion. It is known that up to 10% of disease can be PSMA negative from outset. In addition, the unspecified bone uptake  on PSMA studies is now recognized and MRI is the gold standard in this situations. We have added a paragraph detailing the above.

Round 2

Reviewer 1 Report

Comments and Suggestions for Authors

The authors have addressed my issues and the manuscript quality has been improved. I have no more comments.